# Factors associated with the use of diet and the use of exercise for prostate cancer by long-term survivors

Suzanne Hughes[1], Sam Egger [1]*, Chelsea Carle[1], David P. Smith[1,2,3,4], Suzanne Chambers[4,5], Clare Kahn[1], Cristina M. Caperchione[6], Annette Moxey[7], Dianne L. O'Connell[1,3,7]

1 Cancer Research Division, Cancer Council New South Wales, Sydney, New South Wales, Australia, 2 School of Public Health and Preventive Medicine, Monash University, Melbourne, Australia, 3 School of Public Health, University of Sydney, Sydney, New South Wales, Australia, 4 Menzies Health Institute, Queensland, Griffith University, Gold Coast, Queensland, Australia, 5 Faculty of Health, University of Technology Sydney, Sydney, New South Wales, Australia, 6 Faculty of Health, Human Performance Research Centre, University of Technology Sydney, Sydney, New South Wales, Australia, 7 School of Medicine and Public Health, University of Newcastle, Newcastle, New South Wales, Australia

☯ These authors contributed equally to this work.
* same@nswcc.org.au

**Data Availability Statement:** Data used in this study are partially restricted for use because of restrictions imposed by ethical approval, the research aims governing the study and consent

## Abstract

### Objective

To assess the use of diet and the use of exercise for prostate cancer (and/or its treatments' side effects) by long-term survivors and whether such use is associated with selected socio-demographic, clinical, health-related quality-of-life (HRQOL) and psychological factors.

### Design, setting and participants

Population-based cohort study in New South Wales, Australia of prostate cancer survivors aged <70 years at diagnosis and who returned a 10-year follow-up questionnaire.

### Methods

Validated instruments assessed patient's HRQOL and psychological well-being. Poisson regression was used to estimate adjusted relative proportions (RRs) of prostate cancer survivor groups who were currently eating differently ('using diet') or exercise differently ('using exercise') to help with their prostate cancer.

### Results

996 (61.0% of 1634) participants completed the 10-year questionnaire of whom 118 (11.8%; 95%CI[9.8–13.9]) were using diet and 78 (7.8%; 95%CI[6.2–9.5]) were using exercise to help with their prostate cancer. Men were more likely to use diet or use exercise for prostate cancer if they were younger (p-trend = 0.020 for diet, p-trend = 0.045 for exercise), more educated (p-trend<0.001, p-trend = 0.011), support group participants (p-nominal<0.001, p-nominal = 0.005), had higher Gleason score at diagnosis (p-trend<0.001, p-trend = 0.002)

agreements with study participants. The data is available through an application to the Cancer Council NSW Cancer Research Division and with approval from the Cancer Council NSW Ethics Committee. Researchers who wish to access data owned or held by Cancer Council NSW must hold all relevant ethics approvals before access can be granted. Researchers may be required to enter into a Data Transfer Agreement. To find out how to access data please contact the Cancer Research Division at research@nswcc.org.au. Details and contact information are available at https://www.cancercouncil.com.au/research/for-researchers/ethics/. The data custodian for the NSW Prostate Cancer Care and Outcomes Study is Cancer Council NSW.

**Funding:** This work was supported by the Prostate Cancer Foundation of Australia (PG40) (to DLO'C). DPS was supported by a Career Development Fellowship from the Cancer Institute NSW (#15/CDF/1-10).

**Competing interests:** The authors have declared that no competing interests exist.

and had knowledge of cancer spread (p-nominal = 0.002, p-nominal = 0.001). Use of diet was also associated with receipt of androgen deprivation therapy (RR = 1.59; 95%CI[1.04–2.45]), a greater fear of cancer recurrence (p-trend = 0.010), cognitive avoidance (p-trend = 0.025) and greater perceived control of cancer course (p-trend = 0.014). Use of exercise was also associated with receipt of prostatectomy (RR = 2.02; 95%CI[1.12–3.63]), receipt of androgen deprivation therapy (RR = 2.20; 95%CI[1.34–3.61]) and less satisfaction with medical treatments (p-trend = 0.044).

## Conclusions

Few long-term prostate cancer survivors use diet or exercise to help with their prostate cancer. Survivors may benefit from counselling on the scientific evidence supporting healthy eating and regular exercise for improving quality-of-life and cancer-related outcomes.

## Introduction

Worldwide, prostate cancer is the second most common cancer for men and the fourth most common cancer overall [1]. In higher-income countries where prostate specific antigen (PSA) testing is common, substantial numbers of men are living with a diagnosis of prostate cancer and the adverse consequences (e.g., incontinence, erectile dysfunction, pain, psychological distress) that can persist for many years after diagnosis [2–3]. In New South Wales (NSW), Australia's most populous state, the number of men living with prostate cancer was estimated to have risen by at least 59% (>22,500 extra prevalent cases) in the ten years from 2007 to 2017 [3]. Given the significant and increasing prevalent pool, there is growing interest in lifestyle modifications that might help reduce the risk of prostate cancer progression and improve prostate cancer treatment related quality-of-life [4].

Because poor diet and lack of physical activity are thought to play a role in the development of cancer [5], it seems plausible that these factors may also play a role in the progression of cancer, including prostate cancer. In terms of diet, three randomised controlled trials (RCTs) [6–8]—each judged to be at low risk of bias and of high methodological quality [9]—demonstrated that certain dietary interventions might improve markers of prostate cancer progression. Limited evidence from observational studies has also suggested that engaging in healthy dietary practices (e.g., increased fruit and vegetable consumption, reduced total and saturated fat consumption) may help in reducing prostate cancer progression [10–11]. With regards to exercise, there appears to be little or no evidence from RCTs that changes in physical activity levels slow prostate cancer progression, but evidence of this effect has been found in observational studies [12–14]. For other outcomes, evidence from RCTs suggests that various physical activity interventions can improve physical functioning, physical fitness, muscular strength, body composition and quality of life for patients receiving androgen deprivation therapy (ADT) [15–17]. In addition to the possible benefits of a healthy diet and physical activity on prostate cancer related outcomes, survivors potentially receive additional health benefits unrelated to their cancer, such as reduced risks of cardiovascular disease [18] and additional primary cancers [4].

Taking into account the available evidence and expert clinical opinion, the American Cancer Society Prostate Cancer Survivorship Care Guidelines (ACSSCG) [19] recommend that diets for prostate cancer survivors should—among other things—emphasise vegetables, fruits

and foods with low amounts of saturated fat, while consuming adequate, but not excessive, amounts of dietary sources of calcium. In the absence of physical limitations or contraindications, the ACSSCG recommend that survivors should aim for at least 75 or 150 minutes per week of vigorous or moderate intensity exercise, respectively. The ACSSCG also recommend that survivors with post-prostatectomy incontinence be referred to a physical therapist for pelvic floor rehabilitation.

Despite these recommendations, few previous quantitative studies have examined the dietary and exercise changes that prostate cancer survivors actually adopt to help with their cancer or its treatments' side effects. In this cross-sectional analysis, we used information from a group of long-term survivors of prostate cancer (mean = 10 years) from NSW, to describe the use of diet and the use of exercise "for prostate cancer and/or its treatments' side effects" (hereafter abbreviated as "for prostate cancer") and whether such use was associated with selected socio-demographic, clinical, health-related quality-of-life (HRQOL) and psychological factors. We also discuss the cohort's use of diet and exercise for prostate cancer in comparison to their use of complementary and alternative medicines (CAMs) for prostate cancer—the latter having been reported in detail in a previous publication [20].

## Materials and methods

### Study sample

The New South Wales Prostate Cancer Care and Outcomes Study (PCOS) is a population-wide longitudinal cohort study conducted in NSW, Australia, with a primary objective of assessing the effects of various treatments on HRQOL after a prostate cancer diagnosis. A total of 3195 men aged less than 70 years with histopathologically confirmed T1-4 prostate cancer diagnosed between October 2000 and October 2002 were identified through the NSW Cancer Registry and, after verbal informed consent had been given by their doctor, were invited to participate in PCOS. Of these patients, 1995 completed a baseline questionnaire (S1 Fig) after providing written informed consent. By January 2011, 1634 men were still alive, and 1427 of these men remained in PCOS and were invited to participate in a 10-year follow-up questionnaire (mean of 10 years after diagnosis; range 9–12 years). The 10-year questionnaire assessed various HRQOL and psychological outcomes for participants and their use of diet and exercise for prostate cancer (participants were also asked about their use of CAMs for prostate cancer and the corresponding results were reported in a previous publication [20]). Of the 1634 men, 996 (61%) completed and returned the 10-year questionnaire. Additional details of the initial recruitment process for PCOS are provided elsewhere [21–22]. PCOS was approved by the human research ethics committees of Cancer Council NSW, the Cancer Institute NSW, and the NSW Department of Health. The 10-year follow-up questionnaire was approved by the Cancer Council NSW Human Research Ethics Committee (Approval number: 2010#244).

### Data collection

**Clinical and socio-demographic data.** Clinical data relating to diagnosis and primary treatment were collected for each participant by either a trained field worker or the treating doctor using a data collection form and protocol. These data were collected between 12 and 24 months after the histological diagnosis of prostate cancer and included prostate-specific antigen (PSA) level at diagnosis, Gleason score and clinical stage at diagnosis, and treatment received within 12 months of diagnosis. Place and socio-economic status of each man's residence at diagnosis were based on the Accessibility/Remoteness Index of Australia (ARIA+) [23] and the Socio-Economic Indexes for Areas (SEIFA) [24] respectively. Highest level of education completed was self-reported in the baseline questionnaire. Information on prostate

cancer treatments received was obtained from the treating doctors' records (diagnosis to 12 months) and from linked administrative health datasets (covering from diagnosis to ten-year follow-up). For each man who provided written informed consent, treatment data were obtained from Medicare Australia and NSW Health's Admitted Patient Data Collection [25]. Clinical and socio-demographic information obtained in the 10-year questionnaire included current place of residence, employment status, marital status, support group participation and self-report of whether the cancer had spread.

**Psychological and health-related quality-of-life measures.** A number of previously validated psychological and HRQOL patient-reported outcome measures (PROMs) were included in the 10-year questionnaire including (Table 1): a 6-item course of cancer subscale from the Cancer Locus of Control scale measuring the man's perceived control of the course of their cancer [26]; Kornblith's 5-item Cancer Fear of Recurrence scale [27]; the 22-item Impact of Event Scale-Revised (IES-R) [28] measuring distress, hyperarousal, intrusive thinking and cognitive avoidance associated with having prostate cancer; the 14-item Hospital Anxiety and Depression Scale (HADS) [29] measuring anxiety and depression; the 26-item Expanded Prostate cancer Index Composite Short Form (EPIC-26) [30] measuring urinary incontinence, urinary irritative/obstructive, bowel, sexual and hormonal summary scores, and also measuring urinary, bowel and sexual bother scores from the University of California-Los Angeles Prostate Cancer Index (UCLA-PCI) [31]; and the 12-item Short Form-12 (SF-12) [32] scale measuring

**Table 1. Psychological and health-related quality-of-life patient-reported outcome measures included in the 10-year questionnaire.**

| Patient-reported outcome measure | Domain/scale/subscale |
|---|---|
| Cancer Locus of Control | Perceived control of the course of cancer |
| Kornblith's Fear of Cancer Recurrence | Fear of cancer recurrence |
| IES-R | Prostate cancer specific distress |
| IES-R | Prostate cancer specific hyperarousal |
| IES-R | Prostate cancer specific intrusive thinking |
| IES-R | Prostate cancer specific cognitive avoidance |
| HADS | Anxiety |
| HADS | Depression |
| EPIC-26 | Urinary incontinence summary |
| EPIC-26 | Urinary irritative/obstructive summary |
| EPIC-26 | Bowel summary |
| EPIC-26 | Sexual summary |
| EPIC-26 | Hormonal summary |
| UCLA-PCI | Urinary bother |
| UCLA-PCI | Bowel bother |
| UCLA-PCI | Sexual bother |
| EPIC-50 | Satisfaction with medical treatment |
| SF-12 | Mental component score |
| SF-12 | Physical component score |

EPIC-26 = Expanded Prostate cancer Index Composite Short Form

UCLA-PCI = University of California-Los Angeles Prostate Cancer Index

HADS = Hospital Anxiety and Depression Scale

IES-R = Impact of Event Scale-Revised

EPIC-50 = Expanded Prostate cancer Index Composite Long Form (EPIC-50)

SF-12 = Short Form-12

the mental and physical dimensions of HRQOL. Satisfaction with medical treatments was ascertained on a 1-item 5-point Likert scale from the Expanded Prostate cancer Index Composite Long Form (EPIC-50) [33]. For each psychological domain, higher scores indicate higher levels of the psychological attribute. For each HRQOL domain, higher scores indicate better HRQOL (which corresponds to less bother for the bother domains assessing bother).

**Diet and exercise changes.** Men were asked whether they had "ever" made a change to their diet to help with their prostate cancer and/or its treatments' side effects. If they answered yes they were then asked whether they were "currently" eating differently to help with their prostate cancer and/or its treatments' side effects. Participants were asked about their specific dietary changes with available response options being: increased/decreased fruit, soy products, vegetables, dairy, fats, oils, fried foods, processed meats, red meat, dairy products plus free-text fields for user–specified options. Men were also asked whether they had "ever" made a change to the exercise they do to help with their prostate cancer and/or its treatments' side effects. If they answered yes they were then asked whether they were "currently" exercising differently to help with their prostate cancer and/or its treatments' side effects. Participants were asked to describe any changes in exercise and these free-text fields were subsequently coded for analysis.

## Statistical methods

Poisson regression with robust variance estimation [34] was used to estimate the adjusted relative proportions (RRs) of current diet use and current exercise use for prostate cancer according to socio-demographic characteristics, clinical characteristics, psychological and HRQOL domains. The dependent variables for all regressions were current diet use for prostate cancer (yes or no) and current exercise use for prostate cancer (yes or no). Independent variables included age at completion of the 10-year questionnaire (<65, 65–69, 70–74, 75+ years; age ranged from 52 to 80 years), education (university or college degree, high school, less than high school), socio-economic status of place of residence (divided into quintiles using SEIFA), place of residence (major city, inner regional, outer regional/ remote/ very remote based on ARIA+), health insurance (private health insurance- with extras, private health insurance-without extras, Medicare only), employment status (in full time paid work, in part time paid work, retired/unemployed, self-employed), married or in defacto partner relationship (no, yes), participation in a support group (no contact with support groups, receive newsletter only, participate regularly or occasionally), country of birth (Australia, elsewhere), overall cancer severity at diagnosis (localised low risk, localised intermediate risk, localised high risk, stage T3-4, unknown) [35] and treatments used since diagnosis (active surveillance/watchful waiting (AS/WW), prostatectomy, external beam radiotherapy (EBRT)/ brachytherapy, ADT, bone-EBRT/chemotherapy/bisphosphonates). Other independent variables included PSA level at diagnosis (<4, 4 to <10, 10 to <20, 20+ng/mL, unknown), Gleason score at diagnosis (<7, 7, >7, unknown), clinical stage at diagnosis (T1a-c, T2a-c, T3a-c/T4a, unknown), and knowledge of cancer spread (no, yes). In order to avoid collinearity, however, these other variables were not included in models simultaneously with overall cancer severity at diagnosis. Subjects with missing data on any independent variable were excluded from regression analyses, but 'unknown' test results for clinical variables were analysed as distinct categories (because 'unknown' test results are primarily due to the absence of testing and thus the result is 'unknown' to the subject). Tests for linear trends were performed by inclusion of continuous/ordinal versions of independent variables where appropriate. For ordinal variables that were not interval scaled (eg. health insurance), consecutive integers were used for coding when testing for linear trends. Psychological and HRQOL variables were included one at a time as linear

continuous independent variables in regression models after standardising each variable to have variance equal to one. Hence, RRs for psychological and HRQOL variables indicate the change in the probabilities of using diet or of using exercise per standard deviation increase in the variable [36]. Subjects with missing data for any psychological or HRQOL domain were excluded from analyses relating to that particular domain.

Sensitivity analyses were conducted to assess whether estimates were unduly affected by missing data. In these analyses, multiple imputation was used to impute missing 10-year data for the 638 surviving PCOS participants who did not complete the 10-year questionnaire. Missing data were also multiply imputed for participants who completed the 10-year question-naire but had missing data for one or more variables. Data were imputed 100 times using the method of chained equations [37]. Variables in the imputation models were the dependent and independent variables in the original multivariable regression models plus baseline socio-economic status of place of residence, place of residence, employment status, health insurance and marital status (age, education, and country of birth at the 10-year follow-up were esti-mated directly from corresponding baseline values). The combined imputed and original data were then analysed using estimation techniques for multiply imputed data.

## Results

Of the 996 men who returned a 10-year follow-up questionnaire, 134 (13.5%; 95%CI[11.3–15.6]]) had ever changed their diet and 102 (10.2%; 95%CI[8.4–12.1])) had ever changed the exercise they do to help with their prostate cancer (Table 2). Of the 996 men, 118 (11.8% 95% CI[9.8–13.9]) were currently using diet and 78 (7.8%%; 95%CI[6.2–9.5]) were currently using exercise for their prostate cancer. The most common dietary and exercise changes were the same for 'ever' use and 'current' use including 'more vegetables', 'more fruit', 'less processed

**Table 2. Types of diet and exercise changes ever and currently used for prostate cancer and/or its treatments' side effects.**

|  | Ever | Currently |
|---|---|---|
| Diet and exercise changes for prostate cancer and/or treatments' side effects^ | n (% of 996) | n (% of 996) |
| **Changes to diet, exercise or both** | **193 (19.4)** | **158 (15.9)** |
| **Diet changes:** | **134 (13.5)** | **118 (11.8)** |
| More fruit | 86 (8.6) | 79 (7.9) |
| More soy products | 37 (3.7) | 30 (3.0) |
| More vegetables | 91 (9.1) | 87 (8.7) |
| Less dairy | 49 (4.9) | 48 (4.8) |
| Less fats, oils or fried foods | 61 (6.1) | 60 (6.0) |
| Less processed meats | 71 (7.1) | 71 (7.1) |
| Less red meat | 66 (6.6) | 63 (6.3) |
| **Exercise changes:** | **102 (10.2)** | **78 (7.8)** |
| More walking | 44 (4.4) | 40 (4.0) |
| More aerobic exercise (other than walking) | 15 (1.5) | 13 (1.3) |
| Pelvic floor exercises | 33 (3.3) | 19 (1.9) |
| Resistance exercises | 10 (1.0) | 7 (<1%) |
| Unspecified gym activities* | 18 (1.8) | 13 (1.3) |

Categories of specific diet and exercise changes are listed if ever used by over 1% of men.

^ Men may have more than one type of diet and or exercise change

* Unspecified gym activities may include "more aerobic exercise", "pelvic floor exercises" and/or "resistance exercises" (hence numbers for these categories are likely to be underestimates).

meat', 'more walking', and 'pelvic floor exercises'. A commonly reported reason for both currently using diet and currently using exercise was 'to make me feel better' (45% and 58% of current users respectively) (S1 Table). Current diet users also frequently cited 'to boost my immune system' (46% of current users), 'to prevent cancer returning or spreading' (46%) and 'to do as much as I can for myself' (46%) as reasons. Relatively few of the men using exercise for their prostate cancer (1 in 4) indicated that information about exercise came from their doctor.

Men were more likely to currently use diet or currently use exercise for prostate cancer if they were younger (p-trend = 0.020 for diet, p-trend = 0.045 for exercise), more educated (p-trend<0.001, p-trend = 0.011), support group participants (p-nominal<0.001, p-nominal = 0.005) (Table 3), had higher Gleason score at diagnosis (p-trend<0.001, p-trend = 0.002) (Table 4) or had knowledge of cancer spread (p-nominal = 0.002, p-nominal = 0.001). Men were more likely to currently use exercise if they had received a prostatectomy (RR = 2.02; 95% CI[1.12–3.63]) or ADT (RR = 2.20; 95%CI[1.34–3.61]) (Table 3), and were more likely to currently use diet if they had ever received ADT (RR = 1.59; 95%CI[1.04–2.45]). PSA level was also associated with use of exercise (p-nominal = 0.024), but the difference was driven by the 'unknown' category and not related to clinical levels (Table 4).

For the psychological domains, diet was more likely to be currently used by men who reported a greater fear of cancer recurrence (RR = 1.25; 95%CI[1.05–1.47]), cancer-specific cognitive avoidance (RR = 1.21; 95%CI[1.02–1.42]) and perceived control of cancer course (RR = 1.25; 95%CI[1.05–1.50]) (Fig 1). Current use of exercise was associated with less satisfaction with medical treatments (RR = 0.83; 95%CI[0.69–1.00]) (Fig 1).

Analyses using the multiply-imputed data produced marginally, but not materially, higher estimates for prevalences of current diet and exercise use (S2 Table). The estimated magnitudes of associations from the original analyses were not materially different from those obtained from analyses using the multiply-imputed data (S2–S4 Tables, S2 and S3 Figs).

## Discussion

A cancer diagnosis is often considered to be a 'teachable moment' in which patients may be inclined to seek information about lifestyle changes that might improve cancer outcomes and quality-of-life. In this cohort of long-term prostate cancer survivors, however, only about one in seven men had ever made changes to their diet and only one in ten men had ever made changes to the exercise they do to help with their prostate cancer. In terms of the specific dietary and exercise changes adopted by more than 1% of the cohort (Table 2), most were broadly consistent with recommendations outlined in the ACSSCG [19], and none were unequivocally contraindicated by these guidelines. Younger, more educated survivors and those who participate in support groups were more likely to currently use diet or exercise to help with their prostate cancer. ADT recipients were more likely to currently use diet or exercise than non-recipients, while prostatectomy recipients were more likely to currently use exercise than non-recipients. Psychological distress also appeared to be a motivator for diet and exercise use with current diet users having greater fear of recurrence and cognitive avoidance than non-users, while current exercise users were less satisfied with their medical treatments than non-users.

A number of previous studies have reported on the prevalence of dietary and exercise behaviours among prostate cancer survivors [38–44]. In a 2008 systematic review [45], investigators found that the prevalence of physically active prostate cancer survivors was less than 30% in some study cohorts. With regards to diet, two U.S surveys [38–39] reported that only about 1/3 of prostate cancer survivors were meeting the American Cancer Society's (contemporaneous) recommendations for fruit and vegetable intake [46]. It is important to note that

**Table 3. Associations between current use of diet and exercise for prostate cancer and/or its treatments' side effects and socio-demographic/clinical characteristics for Australian long-term prostate cancer survivors.**

| Characteristic | N | Diet | | | Exercise | | |
|---|---|---|---|---|---|---|---|
| | | Current-user n (%) | RR* | p-nominal p-trend^^ | Current-user n (%) | RR* | p-nominal p-trend^^ |
| | 996 | 118 (11.8) | | | 78 (7.8) | | |
| **Age (years)** | | | | | | | |
| <65 | 164 | 24 (14.6) | ref. | 0.220 | 17 (10.4) | ref. | 0.195 |
| 65–69 | 267 | 37 (13.9) | 1.06 (0.67, 1.67) | 0.020 | 19 (7.1) | 0.70 (0.37, 1.32) | 0.045 |
| 70–74 | 299 | 28 (9.4) | 0.69 (0.41, 1.18) | | 27 (9.0) | 0.88 (0.46, 1.69) | |
| 75–80 | 266 | 29 (10.9) | 0.74 (0.44, 1.24) | | 15 (5.6) | 0.52 (0.26, 1.05) | |
| **Education** | | | | | | | |
| University or college degree | 297 | 57 (19.2) | ref. | <0.001 | 36 (12.1) | ref. | 0.023 |
| High school | 670 | 60 (9.0) | 0.47 (0.32, 0.67) | <0.001 | 42 (6.3) | 0.58 (0.36, 0.93)^ | 0.011 |
| Less than high school | 25 | 1 (4.0) | 0.25 (0.04, 1.53) | | 0 (0.0) | | |
| Missing + | 4 | 0 (0.0) | | | 0 (0.0) | | |
| **Socio-economic status of residence area** | | | | | | | |
| 1- Highest SES | 346 | 46 (13.3) | ref. | 0.438 | 30 (8.7) | ref. | 0.554 |
| 2 | 204 | 24 (11.8) | 1.13 (0.69, 1.83) | 0.914 | 16 (7.8) | 1.30 (0.68, 2.48) | 0.288 |
| 3 | 214 | 21 (9.8) | 0.67 (0.38, 1.17) | | 19 (8.9) | 1.01 (0.52, 1.96) | |
| 4 | 144 | 13 (9.0) | 0.96 (0.48, 1.93) | | 8 (5.6) | 0.67 (0.25, 1.77) | |
| 5- Lowest SES | 83 | 12 (14.5) | 1.24 (0.58, 2.67) | | 4 (4.8) | 0.56 (0.18, 1.77) | |
| Missing + | 5 | 2 (40.0) | | | 1 (20.0) | | |
| **Place of residence** | | | | | | | |
| Major city | 580 | 75 (12.9) | ref. | 0.934 | 50 (8.6) | ref. | 0.237 |
| Inner regional | 266 | 27 (10.2) | 0.91 (0.57, 1.48) | 0.763 | 16 (6.0) | 0.72 (0.38, 1.33) | 0.257 |
| Outer regional/ remote/ very remote | 148 | 15 (10.1) | 0.95 (0.48, 1.88) | | 12 (8.1) | 1.36 (0.66, 2.81) | |
| Missing + | 2 | 1 (50.0) | | | 0 (0.0) | | |
| **Health insurance** | | | | | | | |
| Private—with extras | 584 | 75 (12.8) | ref. | 0.763 | 55 (9.4) | ref. | 0.257 |
| Private—without extras | 153 | 16 (10.5) | 0.81 (0.47, 1.42) | 0.699 | 9 (5.9) | 0.58 (0.28, 1.19) | 0.220 |
| Medicare | 256 | 26 (10.2) | 0.94 (0.58, 1.50) | | 14 (5.5) | 0.73 (0.38, 1.38) | |
| Missing + | 3 | 1 (33.3) | | | 0 (0.0) | | |
| **Employment status** | | | | | | | |
| In full time paid work | 130 | 18 (13.8) | ref. | 0.750 | 9 (6.9) | ref. | 0.359 |
| In part time paid work | 107 | 12 (11.2) | 0.82 (0.41, 1.64) | n/a | 7 (6.5) | 0.94 (0.35, 2.54) | n/a |
| Retired/Unemployed | 740 | 83 (11.2) | 0.98 (0.60, 1.61) | | 59 (8.0) | 1.60 (0.73, 3.53) | |
| Self-employed | 17 | 5 (29.4) | 1.46 (0.53, 4.03) | | 3 (17.6) | 2.34 (0.56, 9.78) | |
| Missing + | 2 | 0 (0.0) | | | 0 (0.0) | | |
| **Married or living with defacto** | | | | | | | |
| No | 166 | 20 (12.0) | ref. | 0.926 | 13 (7.8) | ref. | 0.971 |
| Yes | 829 | 98 (11.8) | 1.02 (0.63, 1.65) | n/a | 65 (7.8) | 1.01 (0.54, 1.89) | n/a |
| Missing + | 1 | 0 (0.0) | | | 0 (0.0) | | |
| **Participate in a support group** | | | | | | | |
| No contact with support groups | 770 | 76 (9.9) | ref. | <0.001 | 46 (6.0) | ref. | 0.005 |
| Receive newsletter only | 153 | 30 (19.6) | 2.13 (1.43, 3.19) | n/a | 20 (13.1) | 2.06 (1.23, 3.45) | n/a |
| Participate regularly or occasionally | 73 | 12 (16.4) | 1.48 (0.85, 2.59) | | 12 (16.4) | 2.09 (1.14, 3.84) | |
| **Country of birth** | | | | | | | |
| In Australia | 768 | 90 (11.7) | ref. | 0.350 | 62 (8.1) | ref. | 0.198 |
| In another country | 227 | 28 (12.3) | 0.82 (0.54, 1.24) | n/a | 16 (7.0) | 0.71 (0.42, 1.20) | n/a |

*(Continued)*

**Table 3.** (Continued)

| Characteristic | N | Diet | | | Exercise | | |
|---|---|---|---|---|---|---|---|
| | | Current-user n (%) | RR* | p-nominal p-trend^^ | Current-user n (%) | RR* | p-nominal p-trend^^ |
| Missing + | 1 | 0 (0.0) | | | 0 (0.0) | | |
| **Overall cancer severity at diagnosis†** | | | | | | | |
| Localised low risk | 341 | 31 (9.1) | ref. | 0.237 | 20 (5.9) | ref. | 0.429 |
| Localised intermediate risk | 359 | 36 (10.0) | 1.13 (0.70, 1.82) | 0.023 | 29 (8.1) | 1.44 (0.83, 2.51) | 0.067 |
| Localised high risk | 176 | 33 (18.8) | 1.56 (0.94, 2.58) | | 16 (9.1) | 1.39 (0.73, 2.64) | |
| Stage T3-4 | 68 | 14 (20.6) | 1.91 (0.99, 3.65) | | 10 (14.7) | 2.17 (0.97, 4.87) | |
| Unknown | 52 | 4 (7.7) | 1.06 (0.39, 2.87) | | 3 (5.8) | 1.27 (0.37, 4.38) | |
| **Treatments used since diagnosis ^^^** | | | | | | | |
| AS/WW | 99 | 13 (13.1) | 1.75 (0.94, 3.26) | <0.001 # | 7 (7.1) | 1.81 (0.86, 3.83) | <0.001 # |
| Prostatectomy | 658 | 65 (9.9) | 0.83 (0.53, 1.30) | n/a | 56 (8.5) | 2.02 (1.12, 3.63) | n/a |
| EBRT/Brachytherapy | 397 | 66 (16.6) | 1.44 (0.91, 2.27) | | 37 (9.3) | 1.17 (0.69, 1.98) | |
| Androgen deprivation therapy | 319 | 60 (18.8) | 1.59 (1.04, 2.45) | | 39 (12.2) | 2.20 (1.34, 3.61) | |
| Other** | 10 | 4 (40.0) | 2.54 (1.04, 6.21) | | 3 (30.0) | 2.58 (0.88, 7.52) | |

* Adjusted for age, education, socio-economic status of residence area, place of residence, health insurance, employment status, marital status, participation in a support group, country of birth, cancer severity at diagnosis, and treatments used since diagnosis;

+ 16 of the 996 participants were excluded from regression analyses due to missing data;

† Localised (stage 1 or 2) risk groups- low risk (PSA≤10, Gleason score ≤6, and clinical stage = T1-2a), intermediate risk (10<PSA≤20, Gleason score = 7 or clinical stage = T2b) high-risk (PSA >20, Gleason score>7, or clinical stage T2c);

p-values values correspond to fully adjusted models;

^Groups merged for regression analysis due to zero cell;

^^p-trend analysis excludes 'missing' and 'unknown' categories;

^^^ Multiple treatments possible for each man and reference group for each treatment is not having had that treatment;

# p-value is for test that all RRs equal one;

** 'Other' group contains 10 patients who received chemotherapy, bisphosphonates and/or bone EBRT;

Data from 10-year questionnaire unless "at diagnosis" stated.

these 'prevalence' studies are only indirectly related to the current study as we did not collect information on the prevalence of specific dietary and exercise behaviours (ie our data relate to 'change' in dietary and exercise behaviours for prostate cancer). The prevalence studies are useful, however, in that they show that some prostate cancer survivorship cohorts have considerable room to improve dietary and exercise behaviours in directions more consistent with survivorship guidelines.

In addition to the 'prevalence' studies mentioned above, another group of studies have reported on 'change' in diet and exercise behaviours following prostate cancer diagnosis, but with 'change' being for any reason and not necessarily related to the cancer or its treatments' side effects [47–48]. The distinction between 'change for any reason' and 'change to help with prostate cancer' is important because individuals–including those who have never been diagnosed with cancer–often make behavioural changes unrelated to cancer. In our study, for example, only 15.9% of survivors reported currently eating, exercising or doing both differently to help with their prostate cancer and/or its treatments' side effects (Table 2), yet 51.4% of survivors reported currently eating, exercising or doing both differently for any reason related to improving health and well-being (including to help with prostate cancer). In the current study, we chose to focus on diet and exercise changes survivors make to help with their prostate cancer because these outcomes capture both men's willingness to make healthy

**Table 4. Associations between current use of diet and exercise for prostate cancer and/or its treatments' side effects and other clinical characteristics for Australian long-term prostate cancer survivors.**

| Characteristic | N+ | Diet | | | Exercise | | |
| --- | --- | --- | --- | --- | --- | --- | --- |
| | | Current-user n (%) | RR* | p-nominal p-trend^^ | Current-user n (%) | RR* | p-nominal p-trend^^ |
| | 996 | 118 (11.8) | | | 78 (7.8) | | |
| **PSA at diagnosis (ng/mL) †** | | | | | | | |
| <10 | 652 | 68 (10.4) | ref. | 0.826 | 49 (7.5) | ref. | 0.024 |
| 10 to <20 | 200 | 28 (14.0) | 1.03 (0.68, 1.58) | 0.333 | 19 (9.5) | 1.30 (0.78, 2.18) | 0.387 |
| 20+ | 97 | 18 (18.6) | 0.79 (0.45, 1.39) | | 7 (7.2) | 0.57 (0.24, 1.38) | |
| Unknown | 47 | 4 (8.5) | 0.78 (0.21, 2.92) | | 3 (6.4) | 0.35 (0.15, 0.82) | |
| **Gleason score at diagnosis †** | | | | | | | |
| <7 | 530 | 45 (8.5) | ref. | 0.013 | 30 (5.7) | ref. | 0.092 |
| 7 | 353 | 48 (13.6) | 1.43 (0.95, 2.14) | <0.001 | 35 (9.9) | 1.43 (0.87, 2.34) | 0.002 |
| >7 | 81 | 22 (27.2) | 2.38 (1.42, 4.00) | | 10 (12.3) | 1.69 (0.85, 3.36) | |
| Unknown | 32 | 3 (9.4) | 1.35 (0.15, 12.37) | | 3 (9.4) | 3.01 (1.06, 8.52) | |
| **Clinical stage at diagnosis †** | | | | | | | |
| T1a-T2a | 709 | 74 (10.4) | ref. | 0.711 | 50 (7.1) | ref. | 0.229 |
| T2b-T2c | 185 | 27 (14.6) | 1.06 (0.68, 1.65) | 0.404 | 15 (8.1) | 0.98 (0.56, 1.70) | 0.350 |
| T3a-c, T4a | 68 | 14 (20.6) | 1.37 (0.78, 2.40) | | 10 (14.7) | 1.59 (0.76, 3.30) | |
| Unknown | 34 | 3 (8.8) | 1.46 (0.22, 9.85) | | 3 (8.8) | 2.30 (0.86, 6.10) | |
| **Knowledge of cancer spread †** | | | | | | | |
| No | 925 | 95 (10.3) | ref. | 0.002 | 59 (6.4) | ref. | 0.001 |
| Yes | 71 | 23 (32.4) | 2.15 (1.31, 3.51) | n/a | 19 (26.8) | 2.53 (1.45, 4.40) | n/a |

* Adjusted for age, education, socio-economic status of residence area, place of residence, health insurance, employment status, marital status, participation in a support group, country of birth, PSA at diagnosis, Gleason score at diagnosis, clinical stage at diagnosis, and treatments used since diagnosis;

† To avoid collinearity, PSA at diagnosis, Gleason score at diagnosis, clinical stage at diagnosis, knowledge of cancer spread were not included in models simultaneously with overall cancer severity at diagnosis;

p-values values correspond to fully adjusted models.

^^p-trend analysis excludes 'missing' and 'unknown' categories;

Data from 10-year questionnaire unless "at diagnosis" stated.

+ 16 of the 996 participants were excluded from regression analyses due to missing data.

lifestyle changes and also their awareness that such changes might help with their prostate cancer. It is important to note, however, that although many survivors in our cohort reported changing eating and exercise habits for reasons unrelated to their cancer, such habits may also help them with their cancer.

Few quantitative studies have assessed the dietary or exercise changes that prostate cancer survivors adopt specifically to help with their cancer. Two studies from the U.S. [49–50] and one from the U.K [51] found similarly small proportions of survivors currently using diet to help with their cancer (12% to 15%) as was observed in the current study (11.8%). In contrast, another study from the U.S [52] reported 27.4% and 15.8% of 114 prostate cancer patients had changed their dietary intake and/or physical activity within the previous 12 months, respectively, to help cope with the cancer or reduce the risk of spread (with changes assessed up to 24 months after diagnosis in 1997 or 1998). Although the estimates from this U.S. study were based on only 114 prostate cancer patients, they are significantly higher than the corresponding 13.5% and 10.2% of survivors in the current study who had 'ever' made dietary and exercise changes for prostate cancer. The study from the U.S. also reported the specific changes in diet and exercise adopted by their cohort and patient factors associated with such changes, but

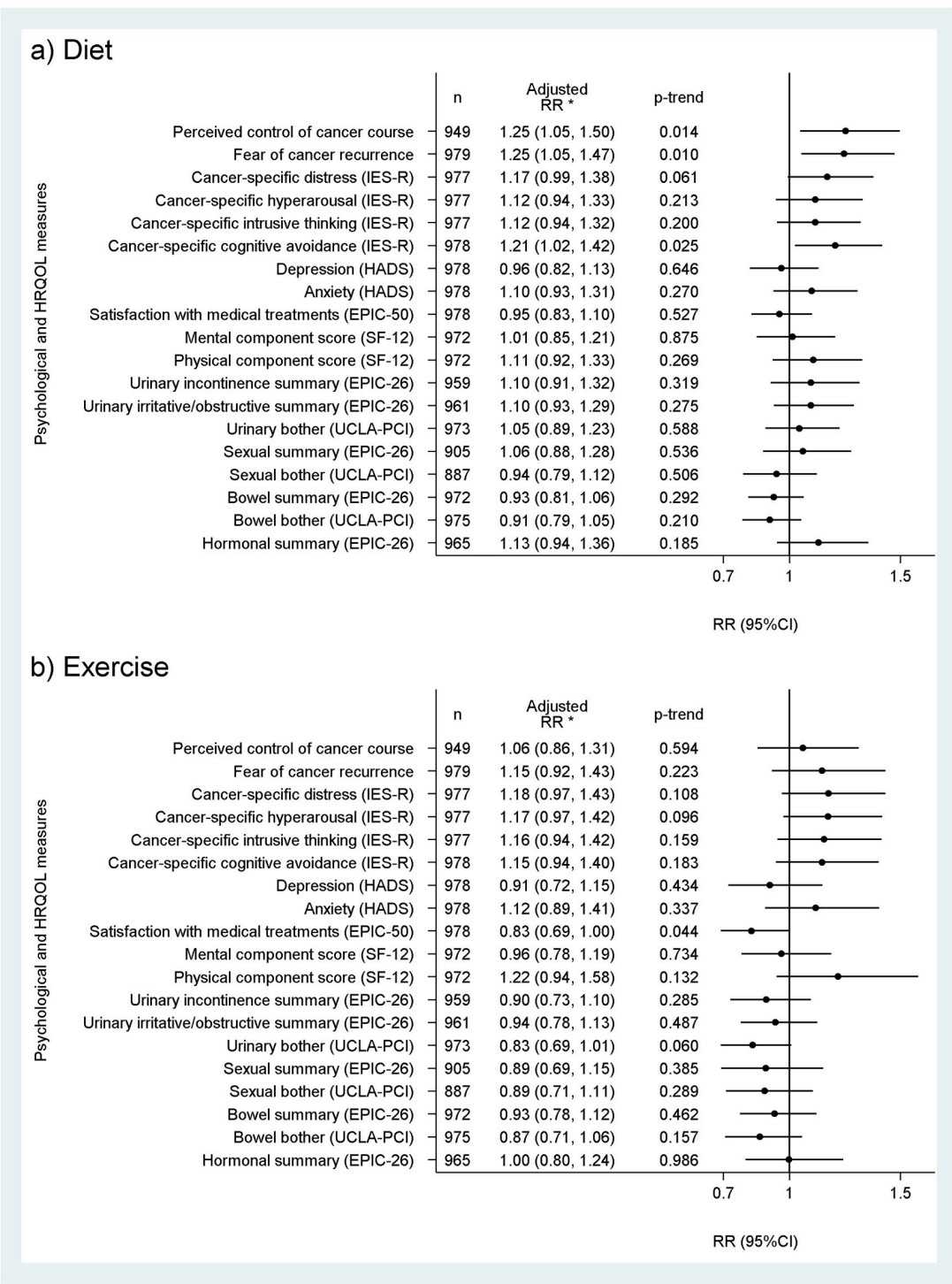

**Fig 1. Relative proportion (RR) of current use of a) diet and b) exercise for prostate cancer and/or its treatments' side effects per standard deviation increase in psychological or HRQOL domain for Australian long-term prostate cancer survivors.** *Adjusted for age, education, socio-economic status of residence area, place of residence, health insurance, employment status, marital status, participation in a support group, country of birth, cancer severity at diagnosis and treatments used since diagnosis.

these results were not reported specifically for prostate cancer survivors (as the cohort also included breast and colon cancer survivors).

In a previous publication, we reported the use of CAMs for prostate cancer by this cohort of 996 long-term survivors [20]. Dietary and exercise modifications and the use of CAMs are similar in that they are activities that many prostate cancer survivors adopt with an aim of improving their cancer and/or treatment related outcomes. These activities differ importantly, however, in that current evidence suggests that prostate cancer survivors might benefit from dietary and exercise modifications, but there is no such evidence supporting the use of CAMs (and some CAMs commonly used for prostate cancer, such as vitamin E [53], have been linked to harmful outcomes). In our cohort, the prevalences of current CAM use [20] and current use of diet, exercise or both for prostate cancer were similar (16.9% versus 15.9%), and exactly half of the 158 survivors currently using diet, exercise or both for their cancer were also using CAMs concurrently for their cancer. Similar psychological factors appear to play a role in survivors' use of CAMs and use of diet for prostate cancer, with both behaviours being related to a fear of cancer recurrence but also seemingly providing men with an increased sense of control over their cancer [20]. Given the high degree of overlapping use of diet, exercise and CAMs for prostate cancer and the presence of common psychological motivators, it may be that survivors who seek advice on CAMs for prostate cancer are likely to be amenable to advice on diet and exercise for prostate cancer.

This study has several limitations. First, potential interactions between the factors associated with the use of diet and exercise for prostate cancer were not examined due to insufficient statistical power. Second, a potentially important source of bias is the non-response of 39% of surviving PCOS participants to the 10-year questionnaire. Somewhat reassuringly, however, is the fact that other than a higher proportion of non-respondents being aged 75 years and over, respondents and non-respondents had similar demographic and clinical characteristics [S5 Table]. Moreover, the inclusion of the 10-year non-respondents through multiple imputation made little difference to the original estimates (S2–S4 Tables, S2 and S3 Figs) or to our overall conclusions. Third, because changes in diet and exercise were based on men's recall measured at one time-point rather than from measurements taken before and after prostate cancer diagnosis, the observed associations cannot be interpreted as causal. Fourth, self-report of having "ever" made changes to diet or exercise for prostate cancer is vulnerable to recall error as men might struggle to remember changes over the 10-year post-diagnosis period. However, self-report of "current" diet or exercise use for prostate cancer would not be vulnerable to the same recall error.

## Conclusion

In this cohort of long-term prostate cancer survivors, to help with their prostate cancer only one in seven prostate cancer survivors had ever made changes to their diet and only one in ten had ever made changes to the exercise they do. Moreover, one in six prostate cancer survivors were currently using diet, exercise or both for their prostate cancer and a similar proportion were currently using CAMs for the same reason. These findings suggest prostate cancer patients may benefit from counselling on the scientific evidence supporting healthy eating and regular exercise and the absence of evidence supporting the use of CAMs. A key challenge, however, is how to motivate men to modify their lifestyles and then maintain these modifications over the course of their (often long) survivorship. Finally, the updating of prostate cancer survivorship guidelines that reference the latest evidence regarding the benefit of diet and exercise for health and wellbeing should be considered a priority in setting the agenda on prostate cancer survivorship.

## Supporting information

**S1 Table. Reasons for—And sources of information on—Current use of diet and exercise for prostate cancer and/or its treatment side effects.**
(DOCX)

**S2 Table. Estimated prevalences of current diet and exercise changes with and without multiple imputation of missing data for surviving participants who completed the baseline but not the 10-year questionnaire, and for participants who completed the 10-year questionnaire but had missing data for one or more variables.**
(DOCX)

**S3 Table. Estimated adjusted relative proportions (RRs) for associations between current diet/exercise changes and socio-demographic/clinical characteristics with and without multiple imputation of missing data for surviving participants who completed the baseline but not the 10-year questionnaire, and for participants who completed the 10-year questionnaire but had missing data for one or more variables.**
(DOCX)

**S4 Table. Comparability of estimated adjusted relative proportions (RRs) corresponding to associations between current diet/exercise changes and other clinical characteristics with and without multiple imputation of missing data for the 638 living participants who completed baseline but not 10-year survey, and for participants who did complete the 10-year questionnaire but had missing data for one or more variables.**
(DOCX)

**S5 Table. Demographic and clinical characteristics of PCOS men invited to participate in the 10-year questionnaire; respondents versus non-respondents.**
(DOCX)

**S1 Fig. Flow diagram showing patients' participation and follow-up.**
(DOCX)

**S2 Fig. Estimated adjusted relative proportions (RRs) for associations between current diet changes and psychological/HRQOL measures with and without multiple imputation of missing data for surviving participants who completed the baseline but not the 10-year questionnaire, and for participants who completed the 10-year questionnaire but had missing data for one or more variables.**
(DOCX)

**S3 Fig. Estimated adjusted relative proportions (RRs) for associations between current exercise changes and psychological/HRQOL measures with and without multiple imputation of missing data for surviving participants who completed the baseline but not 10-year questionnaire, and for participants who completed the 10-year questionnaire but had missing data for one or more variables.**
(DOCX)

## Acknowledgments

We wish to acknowledge the contributions of Carole Pinnock who provided valuable input into the design of the study and analyses of the results, and Maria Albania, who as a research assistant, played a major role in the processes of recruitment, data collection and data entry.

We would also like to thank Olivia Aitkin, Rose Bollard, Alexandra Christian, Brianna Crocker, Peter Gilmore, Richard Hodgkinson, Erica Hodgkinson, Ashoor Khan, Dong Feng Li, Irena Nagorskaia, Caitlin Van der Walt and Lucy Willocquet for their help with data entry, editing and checking. Finally, we would like to thank all the men who participated and made this study possible.

## Author Contributions

**Conceptualization:** Suzanne Hughes, David P. Smith, Suzanne Chambers, Annette Moxey, Dianne L. O'Connell.

**Data curation:** David P. Smith.

**Formal analysis:** Suzanne Hughes, Sam Egger.

**Funding acquisition:** David P. Smith, Dianne L. O'Connell.

**Methodology:** Sam Egger, David P. Smith, Dianne L. O'Connell.

**Supervision:** Dianne L. O'Connell.

**Writing – original draft:** Suzanne Hughes, Sam Egger, Chelsea Carle, David P. Smith, Suzanne Chambers, Clare Kahn, Cristina M. Caperchione, Annette Moxey, Dianne L. O'Connell.

**Writing – review & editing:** Suzanne Hughes, Sam Egger, Chelsea Carle, David P. Smith, Suzanne Chambers, Clare Kahn, Cristina M. Caperchione, Annette Moxey, Dianne L. O'Connell.

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
