## [Decision Letter · Decision Letter 0]

7 Aug 2019

PONE-D-19-20347

Factors associated with the use of diet and the use of exercise for prostate cancer by long-term survivors

PLOS ONE

Dear Mr Egger,

Thank you for submitting your manuscript to PLOS ONE. After careful consideration, we feel that it has merit but does not fully meet PLOS ONE’s publication criteria as it currently stands. Therefore, we invite you to submit a revised version of the manuscript that addresses the points raised during the review process.

This manuscript has been evaluated by two referees with expertise in oncology, epidemiology, nutrition, and exercise science. The referees identified several important weaknesses that should be prioritized in the revised manuscript. Reviewer 1 identified multiple major sources of bias resulting from the sampling process and the recall interval. Please think carefully about how to address this important issue and the validity of your study conclusions. Reviewer 2 identified several limitations as they relate to the questionnaires and their inability to parse out specific types of physical activity. Please consider how the wording of your questionnaire may influence your study results and the specificity of recommendations that can be offered to patients, healthcare providers, and policymakers.

We would appreciate receiving your revised manuscript by Sep 21 2019 11:59PM. To enhance the reproducibility of your results, we recommend that if applicable you deposit your laboratory protocols in protocols.io, where a protocol can be assigned its own identifier (DOI) such that it can be cited independently in the future. For instructions see: http://journals.plos.org/plosone/s/submission-guidelines#loc-laboratory-protocols

We look forward to receiving your revised manuscript.

Kind regards,

Justin C. Brown

Academic Editor

PLOS ONE

Journal Requirements:

2. In the Methods, please clarify whether participants provided informed consent to participate, and if so whether consent was written or verbal.

[No].

6. Please amend either the title on the online submission form (via Edit Submission) or the title in the manuscript so that they are identical.

Reviewers' comments:

Reviewer's Responses to Questions

**Comments to the Author**

1. Is the manuscript technically sound, and do the data support the conclusions?

Reviewer #1: Partly

Reviewer #2: Partly

2. Has the statistical analysis been performed appropriately and rigorously? 

Reviewer #1: Yes

Reviewer #2: Yes

3. Have the authors made all data underlying the findings in their manuscript fully available?

Reviewer #1: No

Reviewer #2: Yes

4. Is the manuscript presented in an intelligible fashion and written in standard English?

Reviewer #1: Yes

Reviewer #2: Yes

5. Review Comments to the Author

Reviewer #1: In this cross-sectional analysis, the investigators aimed to (1) describe the prevalence of self-reported diet and exercise changes among prostate cancer survivors and (2) examine associations with sociodemographic, clinical, HRQOL, and psychological factors. While these are clinically important aims that could help to inform targeted lifestyle interventions among men diagnosed with prostate cancer, the significance and soundness of this study are limited by its design.

Major comments:

• Selection bias is likely given that the analyses were restricted to individuals who survived and remained under follow-up long enough to return a 10-year follow-up questionnaire.

• Measurement bias is likely as men recalled whether they had ever changed certain behaviors (e.g., increased fruit intake, increased walking) over the past 10-year period.

Minor comments:

• The composite measures of “change in diet and exercise” do not capture information on specific behaviors, which would be important to inform targeted interventions.

• The restriction to “change to help with prostate cancer” does not capture positive changes that may be made to benefit other clinically relevant outcomes in this patient population, such as cardiovascular disease.

• The concluding remarks about CAM do not seem central to the a priori aims.

Reviewer #2: (Introduction, Page 4): “There appears to be little or no evidence from RCT’s that changes in physical activity levels improve markers of prostate cancer progression” – This might be true, but there is epidemiological evidence to this effect, pertaining to prostate cancer progression / aggression, and so too post-diagnosis physical activity and overall survival. It may be at-least worthwhile to highlight this, as the working hypothesis isn’t narrowcast to quality of life only – which use patient self-report much like your own current study, thus as dutiful I would think:

https://www.ncbi.nlm.nih.gov/pubmed/26276753

https://www.ncbi.nlm.nih.gov/pubmed/21610110

https://www.ncbi.nlm.nih.gov/pubmed/21205749

(Methods, Page 8, Diet and Exercise Changes): Unsure why “or exercise program in a separate question” is bracketed. It is not a lower-order question, and should simply be included in the statement / sentence. Seems to be a second-order issue with the authors more interested in diet. Treat both equally given the title of the paper and outcomes of the paper aim for this. Beyond this, is the question: “have you ever made a change” really sensitive enough? Seems remarkably general. Of interest would be “how long” they upheld the change, because health behaviour change is complex, and sustaining a behaviour change while undergoing treatment is even more difficult that simply indicating whether “a change” was transiently made. Was this asked or measured? Ifso consider adding to the paper as it will have meaning towards behaviour change coaching or plans in this population too.

(Methods, Page 9, Statistical Methods): It is outlined here that Independent Variables include Age - <65, 65-69, 70-75, 75+ with the age ranging from 52 to 80 – but in your methods and your abstract, you stated men <70 years…. Can you please explain this? Were the Men <70 years that age at recruitment? Thus 10 year follow-up can be up to 80 years of age? Or did this study only include analysis of men currently 70 years or younger? In either case, this might need to be more clearly detailed so as to be less ambiguous. Was there any reason that you decided to exclude subjects with missing data from psychological/HRQOL domains instead of using missing data imputations? Did this apply for other data or variables?

(Table 2): I am surprised there is no information around resistance exercise. Why was walking judged to be different to aerobic exercise? And why has no resistance exercise or muscle building exercise been included? Did the questionnaire assist patients in understanding how different types of exercise or physical activity are defined? If not this is a limitation that must be noted, because participants often miscategorise activities as they do not understand the true definition – it is noticeable in the Godin Leisure-Time Questionnaire often. The exercise portion as a result seems incomplete in this Table – though this may be a consequence of the fact that it was coded by the researchers using free-form entry by participants only.

(Discussion, Page 15): It discusses prevalence studies in the 2nd paragraph – yet it seems to have missed this paper: https://www.ncbi.nlm.nih.gov/pubmed/27647712 which highlights this type of information pertaining to aerobic guidelines and prostate cancer patients with bone metastases. This might be relevant. Also, why was the “data not shown” for the Men who listed “changing exercise or diet for any reason”. Some of these Men may have changed these factors following their prostate cancer diagnosis as a catalyst for the change even if they do not directly attribute to using it “as medicine”. That is, the Men might not have identified their decision to change exercise or diet habits was to “treat their cancer or side-effects” but they may have changed exercise or diet habits “to be more healthy” following their cancer diagnosis. So, you might be removing legitimately interesting data simply because Men didn’t narrowcast their own responses to a self-report questionnaire 10 years post-diagnosis. This may also need to be a limitation acknowledged in your paper. Men might have improved their lifestyle practices, noting that general health and wellbeing is important, not recognising that actually improving general health and wellbeing while dealing with cancer and treatment will itself help “with their prostate cancer”…. This is a flaw of the questionnaire not clearly being able to delineate between the nuance of this, and not the participants themselves. So, this definitely needs some level of appreciation in your paper, otherwise it makes the situation look far direr than it probably is.

6. PLOS authors have the option to publish the peer review history of their article (what does this mean?). If published, this will include your full peer review and any attached files.

Reviewer #1: No

Reviewer #2: No

---

## [Author Response · Author response to Decision Letter 0]

19 Sep 2019

Please see "Reponse to Reviewers" document.

---

## [Editor Report · Decision Letter 1]

23 Sep 2019

Factors associated with the use of diet and the use of exercise for prostate cancer by long-term survivors

PONE-D-19-20347R1

Dear Dr. Egger,

We are pleased to inform you that your manuscript has been judged scientifically suitable for publication and will be formally accepted for publication once it complies with all outstanding technical requirements.

With kind regards,

Justin C. Brown

Academic Editor

PLOS ONE